# Reusable Macroporous Oil Sorbent Films from Plastic Wastes

**DOI:** 10.3390/polym14224867

**Published:** 2022-11-11

**Authors:** Junaid Saleem, Moghal Zubair Khalid Baig, Adriaan Stephanus Luyt, Rana Abdul Shakoor, Said Mansour, Gordon McKay

**Affiliations:** 1Division of Sustainable Development, College of Science and Engineering, Hamad Bin Khalifa University, Doha P.O. Box 34110, Qatar; 2Center for Advanced Materials, Qatar University, Doha P.O. Box 2713, Qatar; 3Qatar Environment and Energy Research Institute, Hamad Bin Khalifa University, Doha P.O. Box 34110, Qatar

**Keywords:** plastic waste, green polymer science, oil sorbent, recycling, upcycling, porous

## Abstract

Plastic waste comprises 15% of the total municipal solid waste and can be a rich source for producing value-added materials. Among them, polyethylene (PE) and polypropylene (PP) account for 60% of the total plastic waste, mainly due to their low-end and one-time-use applications. Herein, we report reusable oil sorbent films made by upcycling waste PE and PP. The as-prepared oil sorbent had an uptake capacity of 55 g/g. SEM analysis revealed a macroporous structure with a pore size range of 1–10 µm, which facilitates oil sorption. Similarly, the contact angle values reflected the oleophilic nature of the sorbent. Moreover, thermal properties and crystallinity were examined using DSC, while mechanical properties were calculated using tensile testing. Lastly, 95% of the sorbed oil could be easily recovered by squeezing mechanically or manually.

## 1. Introduction

Plastic is a major component of municipal solid wastes, with a growing emphasis on its manufacturing size and environmental impact [1,2,3]. A large volume of plastic waste is thrown into the ocean, which creates further problems as around 13 million tons of plastic is ingested by seabirds and fish [4,5,6]. In addition, it is estimated that around 380 million tons of plastic are manufactured every year, and less than 20% is recycled. With this rate, it is projected that by 2050, the oceans will contain more plastic by weight than fish if proper recycling measures are not taken [7].

Polyethylene (PE) and polypropylene (PP) account for 60% of the total plastic waste [8,9]. They are two of the most prevalent polymers, thanks to their lightweight, low cost, and processability [1,10]. A few examples of their low-end and one-time-use applications are sorbent sheets for oil and organic removal, as a support to sanitary pads and diapers, and as disposable plastic bags. However, they are difficult to be separated quantitatively due to their similar densities [11] and required high-energy intensive separation techniques, such as triboelectric separation [12,13], magnetic density separation [14,15], laser-induced breakdown spectroscopy [16], and hyperspectral imaging [17].

Hence, researchers have used recycled PE and PP to produce blended pellets with improved flowability, processability, and compatibility [11,18,19,20,21,22,23]. Inorganic or organic fillers, such as talc, clay, glass fibers, and chalk, have also been added as reinforcement agents [18,22,23].

Another way to use plastic wastes is to convert them into value-added products, such as oil sorbent sheets for oil spill control and prevention. Several thin-film sorbents with micropores have been reported using polyethylene and polypropylene [24,25,26,27,28,29,30]. Oil sorbents based on polymers [31] and plastics [32,33] have been reported for oil water separation and oil spill recovery.

Moreover, commercial sorbent pads are also made of nonwoven polypropylene fabric. They are based on micro-sized individual fibers to form a flexible solid fabric thin film of 10 to 30 µm. These films are stacked to make a thick sorbent pad, as in the case of 3M-HP-255, 3M-156, Chemtex-BP-9W, and Alsorb. Since they float below the oil surface due to their thickness of at least 5 mm, they cannot be used effectively in thin water-borne oil films. In addition to the limited oil capacity of these pads, they are useful for one-time application, thus causing further pollution. Thus, it is imperative to prepare an oil sorbent that is not only capable of absorbing large quantities of oil but also reusable. We present an innovative approach for addressing two environmental issues simultaneously by converting plastic waste into reusable oil sorbents.

## 2. Experimental Section

### 2.1. Method and Materials

Briefly, 3 g of isolated plastic waste comprising equal amounts of PE (in the form of high-density polyethylene) and PP was taken in a round-bottomed flask. Next, 50 mL of *o*-xylene (other forms of xylene can also be used) was added. The reaction mixture was heated to 130 °C until a clear solution was obtained. Usually, the polymer dissolves in 15–30 min. The round-bottomed flask was connected to a reflux condenser to avoid solvent loss. The solution was then poured onto a glass substrate, which was heated to 120 °C and placed on a customized spin coater chuck. Spin coating was performed at 400 rpm for 5 s, then at 1000 rpm for 60 s, and then at 3000 rpm for 60 s to achieve a thin film with a thickness of 20 mm and a porosity of 78%. The film was then heated in a hot-air oven at 150 °C for 20 min, and it was peeled off using a tweezer to obtain an oil sorbent film.

Waste milk bottles from Baladna, Qatar, and ice-cream buckets from Dandy were used as HDPE and PP, respectively. In addition, *o*-xylene, toluene, and paraffin oil from Sigma-Aldrich were used as received without further purification. Engine oil SAE 20W50 and other oils were purchased from a local vendor. A spin coater from Ossila was used for spin coating. PerkinElmer DSC 8500 was used for differential scanning calorimetry. A friction/peel tester (Lloyd Instruments Ltd. Bognor Regis, UK) was used for tensile strength and modulus calculations. The optical contact angle was calculated using OCA 35 (Dataphysics Instruments GmbH, Filderstadt, Germany). SEM images were captured using FEI Quanta650FEG.

### 2.2. Characterizations

Specimens 5 cm in size with an area of 25 cm^2^ were taken and investigated for oil sorption studies. They were weighed individually, and the sorbent was placed in the sorbate. In the saturation studies, the sorbents were placed on the sorbate for 10, 20, 30, and 60 s and were then taken out using a tweezer and allowed to hang for a specific duration so that the loosely connected sorbate was drained until equilibrium was reached. The saturation capacity was then measured. In the dripping kinetic, the sorbent was placed in the sorbate until saturation was reached and then allowed to hang for different times, i.e., 0.5, 1, 2, 5, and 15 min. SEM was performed to assess the surface morphology of the sorbent film. Thermal properties were investigated using DSC, while mechanical properties were determined using tensile testing. Lastly, oleophilic characteristics were confirmed by contact angle measurements.

## 3. Results and Discussion

### 3.1. SEM

The scanning electron microscope image of a macroporous oil sorbent film (Figure 1a) showed a highly porous and weak structure with loosely connected fibrils. Despite 78% porosity, it was not sufficiently strong to be peeled off and used as an oil sorbent on its own. Hence, to improve the strength of the sorbent film, it was heated in a hot-air oven, which provided the necessary strength to the internal structure due to the rearrangement and compact alignment of the polymer chains. Figure 1b reveals the mesh-like structure with interconnected fibrils, confirming homogeneity and better compatibility between HDPE and PP chains with a pore size ranging from 1 to 10 µm. The macropores facilitate the oil sorption mechanism as they create a capillary action that holds oil molecules and prevents them from dripping out. Second, oil molecules within macropores attract like molecules on the surface of the sorbent due to cohesion. Third, adhesion between the oleophilic surface of the sorbent and oil molecules further facilitates oil retention. The combination of cohesion, adhesion, and sorption leads to higher uptake capacity.

### 3.2. Oil Study

Figure 2a presents the saturation kinetics study between the sorbent film and the engine oil. Noticeably, the sorbent film reached its saturation within 30 s. The super-fast oil uptake is especially useful for thin water-borne oil layers as it prevents the spreading of oil from the spill site. Second, since most commercial sorbents are at least 5 mm thick, part of them remains inside the water, especially on sites where a thin layer of oil is to be removed, thereby absorbing unnecessary water.

Figure 2b shows the dripping profile of the sorbent film. When the sorbent film is allowed to hang after removing from the oil bath, the loosely connected oil drips off the sorbent. This process continues until equilibrium is reached, where no more oil escapes from the sorbent. At this point, the true retention capacity can be calculated. In our case, the sorbent film reached its equilibrium capacity within 5 min of dripping and further dripping was not observed. This aspect is particularly useful to differentiate between oil uptake and retention capacities.

**Figure 2 polymers-14-04867-f002:**
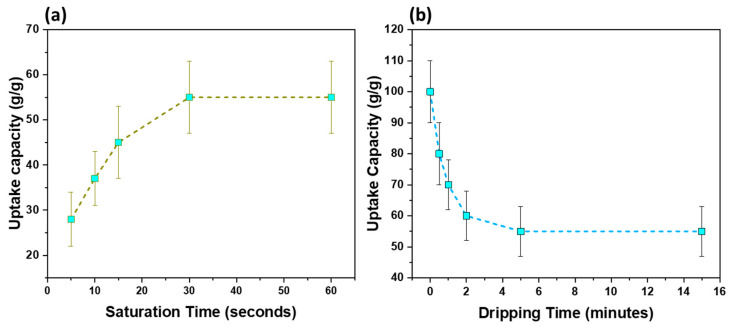
Oil sorption studies of the oil sorbent film using engine oil (density 0.89 g/cm^3^): (**a**) saturation kinetics and (**b**) dripping kinetics.

### 3.3. Recyclability and Contact Angle

Figure 3a presents an uptake comparison using different sorbates with different densities (see Table 1). The highest equilibrium uptake value for the sorbent was 60 g/g using crude oil, and the lowest was 15 g/g using toluene. The reusability study of the sorbent film is shown in Figure 3b. Squeezing the sorbent film mechanically or with bare hands yields 95% sorbate; therefore, the sorbent can be reused as many times as required with 95% oil sorption efficiency. We did not observe wear and tear of the sorbent even after multiple cycles, through which we anticipate no changes in the polymer molecular weight. Contact angles with water, engine oil, and toluene are presented in Figure 3c, which confirms the hydrophobic and oleophilic nature of the sorbent film. When toluene contacted the surface, the liquid was spread in a fraction of a second, thus resulting in a contact angle of less than 1°.

### 3.4. DSC and Tensile Strength

DSC was carried out to investigate thermal properties, enthalpy change, and relative percentage crystallinity, as presented in Figure 4a. The degree of crystallinity can be determined by comparing the change in enthalpy of the polymer sample to that of a 100% crystal [30]. The enthalpy change values for pure crystal PE and PP are 293 and 207 J/g, respectively [34,35]. The area under the curve of PE and PP gives the enthalpy value, which reveals the amount of heat required per milligram of a certain polymer that undergoes a phase change from solid to molten liquid. Usually, crystalline molecules require more energy and heat to melt as the molecules are more compact and denser. It was observed that the sorbent film collected after spin coating was amorphous in nature, with loosely interconnected fibrils; therefore, it could not hold its structure without a support or substrate. There was a lower enthalpy change (112.3 J/g), resulting in a lower percentage of crystallinity (44.9%). Hence, we need to add another step to strengthen the thin-film sorbent by heating it to a higher temperature. Various experiments were conducted to find out the optimum heating temperature to attain optimized porosity and strength, as shown in Table 2. Upon heating, the polymer chains become soft and the space between them reduces, thereby strengthening the intermolecular dispersion forces. This enhancement in dispersion forces realigns and rearranges the polymer chains to more compact and denser chains, resulting in an increased enthalpy change (139.1 J/g) and crystallinity (55.6%), as presented in Table 3 and Table 4. The heating of the thin film facilitated easy separation of the film from the substrate, and the resultant thin film possesses sufficient strength to hold its structural integrity and can be used as a freestanding thin film. The observation is that upon gradual heating of the polymer, the internal fractures at the nano-level start to heal. A mesh-like structure was observed with some rough edges. The edges smoothened with further heating, and the polymer started aggregating as it melted, resulting in macropore formation. Additionally, it was found that heating the samples enhances their tensile strength (see Figure 4b) at the cost of decreasing porosity. Based on the data presented in Table 2, a porosity of 33% and a strength of 8 MPa were found to be a better trade- off for an oil sorbent film with sufficient strength. We further checked the tensile strength of the sorbent film after oil removal through a hexane wash, which gave a similar result of 8 MPa even after multiple cycles, which suggests the robust nature of the sorbent film.

## 4. Conclusions

The approach of fabricating oil sorbents from plastic waste combines an effective way to address two environmental issues simultaneously. The equilibrium oil uptake capacity was found to be 55 g/g, though it varies with the type of oil. Oil saturation is reached within 30 s of contact with the sorbent. The dripping profile reflects the true retention capacity of the sorbent. SEM analysis revealed a macroporous structure composed of interconnected fibrils, with pores ranging from 1 to 10 µm, facilitating oil sorption. DSC and tensile testing revealed the semi-crystalline nature of the sorbent with sufficient mechanical strength to be used in all types of practical applications. Lastly, the sorbent can be reused with 95% efficiency, and oil can be recovered easily by squeezing with bare hands. Combining these attributes with recycled material, the as-prepared sorbent meets the requirements of an effective oil sorbent.

## Figures and Tables

**Figure 1 polymers-14-04867-f001:**
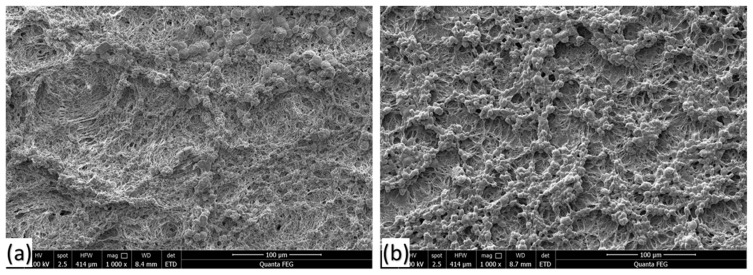
SEM micrographs showing a mesh-like macroporous structure (**a**) before heating and (**b**) after heating the thin-film sorbent.

**Figure 3 polymers-14-04867-f003:**
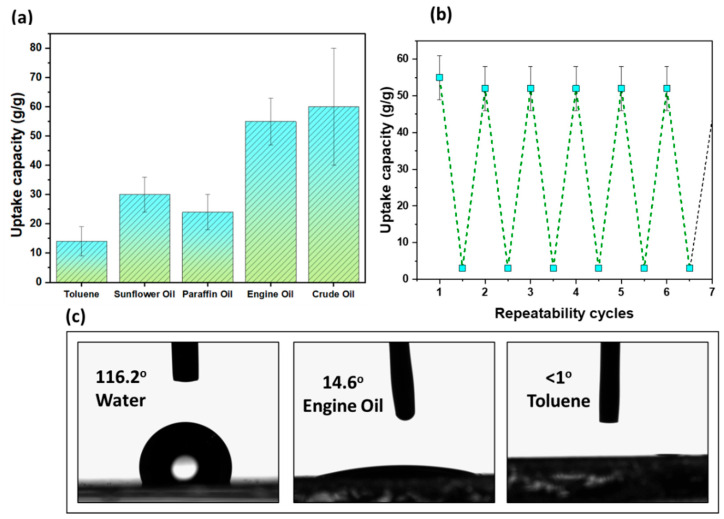
(**a**) Uptake capacity of the sorbent film with different sorbates, (**b**) recyclability of the sorbent film, and (**c**) contact angles with different liquids.

**Figure 4 polymers-14-04867-f004:**
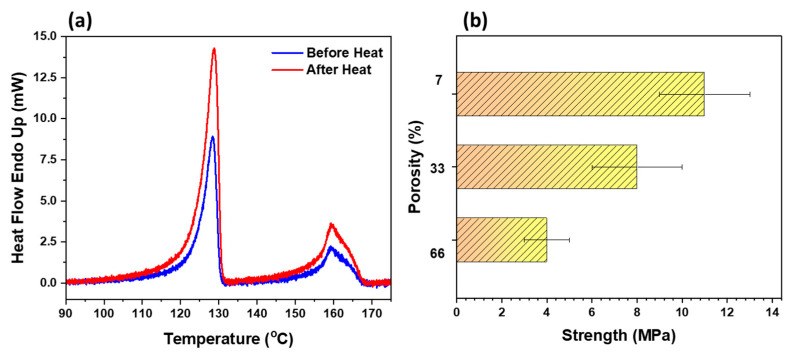
(**a**) DSC spectra before heating and after heating the film and (**b**) porosity vs. strength of the oil sorbent film.

**Table 1 polymers-14-04867-t001:** Densities of the sorbates.

Sorbate	Density (kg/m^3^)
Toluene	867.0
Sunflower oil	918.8
Paraffin oil	800.0
Crude oil	890.0
Engine oil	890.0

**Table 2 polymers-14-04867-t002:** Study of various parameters to optimize the oil sorbent film.

SN	Porosity (%)	Temperature (°C)	Time (min)	Strength (MPa)
1	78	25	0	^1^ ND
2	73	150	5	ND
3	67	150	10	1
4	61	150	15	4
5	33	150	20	8
6	7	150	25	11
7	<1	165	5	12

^1^ Not determined.

**Table 3 polymers-14-04867-t003:** Enthalpy change before and after heating the oil sorbent film.

Polymer	Enthalpy Change (J/g)	
	Before Heat	After Heat
HDPE	83.8	91.4
PP	28.5	47.7
HDPE-PP	112.3	139.1

**Table 4 polymers-14-04867-t004:** Relative percentage crystallinity before and after heating the oil sorbent film using DSC data.

Polymer	% Relative Crystallinity		% Change in Relative Crystallinity
	Before Heat	After Heat	
HDPE	57.2	62.4	5.2
PP	27.5	46.1	18.6
HDPE-PP	44.9	55.6	11.5

## Data Availability

Not applicable.

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
