# Peer review of "Reusable Macroporous Oil Sorbent Films from Plastic Wastes"

_polymers, 2022, doi:10.3390/polym14224867_

Round 1

Reviewer 1 Report

Comments are in attached file.

Reviewer 2 Report

1. The effect of polymer molecular weight on the reusability of the adsorbents should be discussed.

2. The mechanical strength of the polymer materials should be discussed in details, quantitative data is necessary. The application as well as the regeneration requires materials with excellent mechanical stability.

3. Examples for the non-chemical upcycling of PE and PP into film materials should be briefly mentioned in the introduction, in particular the approaches that are more sustainable.

4. The results on the enthalpy change before and after heating for oil sorbent film should be better explained in the text. This part is ambiguous and in-depth discussions are missing.

5. The source and grade/purity of all the sorbates including toluene, sunflower oil, paraffin oil, synthetic oil, enginer oil should be listed in the Experimental section.

6. SEM images show the surface of the prepared films, however cross-section SEM images are missing. It is important to show before- and after-oil-adsorption cross-sectional images to understand the morphology of the studied films, and reveal any changes.

7. Polymer oil sorbents, in particular from plastic waste, are emerging and should be acknowledged (10.1021/acs.iecr.2c01431; 10.1016/j.susmat.2021.e00268; 10.1016/j.cej.2022.137821; 10.1016/j.jece.2021.107105).

8. Reproducibility should be demonstrated. Add error bars as standard deviations on the quantitative data to show the performance, for instance for the uptake capacity in Figures 2 and 3, as well as the porosity in Figure 4.

9. Crude oil should also be studied. Most of the oil spills are crude oil and not the selected sorbates.

10. The conclusions are somewhat vague and general. Provide statements on the main findings and use the most important values to support the statements. How has the field been advanced? Add more clarification.

Round 2

Reviewer 2 Report

The comments got addressed, manuscript can be published.